# SCALING LAWS FOR PRE-TRAINING AGENTS AND WORLD MODELS

## ABSTRACT

The performance of embodied agents has been shown to improve by increasing model parameters, dataset size, and compute. This has been demonstrated in domains from robotics to video games, when generative learning objectives on offline datasets (pre-training) are used to model an agent's behavior (imitation learning) or their environment (world modeling). This paper characterizes the role of scale in these tasks more precisely. Going beyond the simple intuition that 'bigger is better', we show that the same types of power laws found in language modeling (e.g. between loss and optimal model size), also arise in world modeling and imitation learning. However, the coefficients of these laws are heavily influenced by the tokenizer, task & architecture – this has important implications on the optimal sizing of models and data.

## 1 INTRODUCTION

Much progress in AI in the early 2020's has been driven by increasing model size, dataset size, and training compute. Whilst conceptually simple, the importance of this practice has led to an emerging subfield studying the *science of scaling*. This field answers questions such as how to estimate the benefit of increased compute investment, or how to optimally trade-off model and dataset size.

The role of scale in pre-training is until now best understood in the context of large language models (LLMs). Following the observation that the empirical relationship between loss and key scaling quantities can be accurately described by power laws (Kaplan et al., 2020), ensuing work studied the precise trade-off between model and dataset size (Hoffmann et al., 2022), as well as considerations about inference compute (Sardana and Frankle, 2023), repeated training data (Muennighoff et al., 2024), parameter counting (Pearce and Song, 2024), and more (Section 2).

In comparison, less is understood about scaling in *embodied AI*. Recent high-impact works suggest increasing model and dataset size can lead to ever more capable agents for two pre-training objectives; behavior cloning (BC) (Reed et al., 2022; Baker et al., 2022; Brohan et al., 2023) and world modeling (Hafner et al., 2020; Hu et al., 2023; Yang et al., 2023; Bruce et al., 2024). Such works typically demonstrate the benefit of scale through ablations over only a few model sizes, shown in terms of downstream agent performance, confirming the intuition that 'bigger is better' (Sartor and Thompson (2024) provide an aggregated analysis). However, this leaves a large gap to the precise understanding of scale in LLMs, where for a given increase in compute, models can be sized optimally, and their expected performance accurately predicted.

This paper helps close this gap. Similar to the initial study of scale in LLMs, we focus on the effect of scaling on a *generative pre-training loss* (rather than on downstream agent performance, or reward- or representation-centric objectives), in the infinite data regime, on a fixed offline dataset. Under this setting, we train families of transformers on next-token prediction tasks using architectures popular in both world modeling and BC tasks. This leads to several contributions, summarized in Figure 1.

- Scaling laws similar to those in LLMs can be observed in world modeling with tokenized observations and actions (Section 4.1, Figure 1a).

- The optimal trade-off between model and dataset size in world modeling is influenced by the tokenizer's compression rate (number of tokens per observation) (Section 4.1, Figure 1a & b).

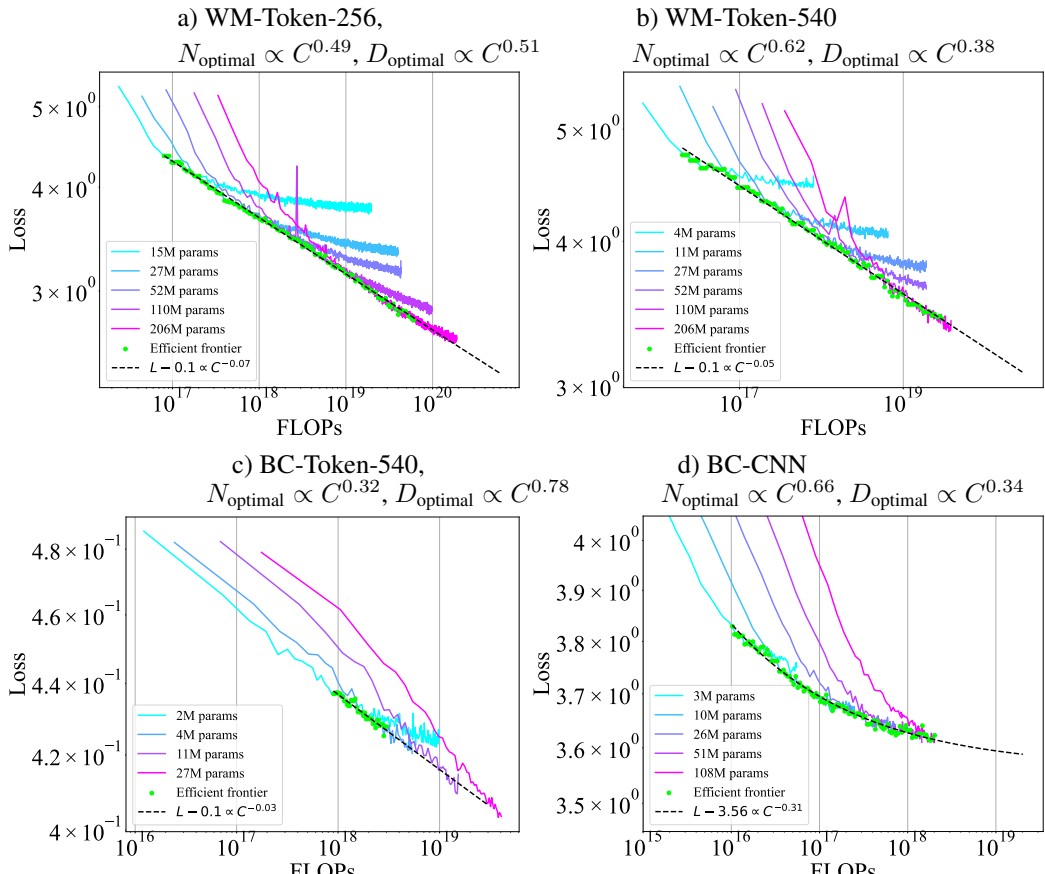

Figure 1: This paper observes that scaling laws, as originally found in LLMs, also emerge in the tasks of world modeling and BC, when studying the pre-training loss on large datasets of human behavior. **a, b)** For world modeling, the power law coefficient determining optimal model size is affected by the compression rate of the tokenizer. **c)** In BC with tokenized image observations (BC-Token), small models need a large FLOPs budget to saturate, making these scaling laws less clear cut. **d)** However, moving to a single continuous embedding per observation remedies this (BC-CNN), producing prototypical scaling laws and a more balanced optimal model size coefficient.

- Scaling laws for BC with tokenized observations are harder to observe under modest compute budgets. The optimal trade-off favors smaller models and more data (Section 4.2, Figure 1c).

- Scaling laws similar to those in LLMs can once again be observed in BC with one continuous encoding per observation (Section 4.2, Figure 1d).

- Our findings can be understood through small-scale language modeling experiments (Section 5).

**Organization.** Section 2 provides detailed related work, contrasting the current understanding of scaling in embodied AI with other domains. Section 3 introduces details for our main experiments, including the architectures considered and details of scaling laws analyses. Section 4.1 & 4.2 present our main results in world modeling and BC respectively. Section 5 presents insights behind our main results, including a set of tiny-scale language experiments mimicking aspects of our main experiments. Section 6 discusses our findings and notes limitations in our work.

## 2 RELATED WORK

**Scaling laws origin.** The term *scaling laws* is used throughout the engineering and physical sciences to denote power law relationships between two quantities, e.g. duration of a volcanic eruption and the probability of it continuing (Cannavò and Nunnari, 2016). The name derives from the *scale-invariant*[1] property of power laws. While early work suggested that power laws could be good empirical descriptors of important variables in deep learning (Hestness et al., 2017; Rosenfeld et al., 2019), it was Kaplan et al. (2020) who provided a comprehensive study of power laws in transformer LLMs, and popularized the usage of *scaling laws* in this context.

**Scaling laws in LLMs.** As the real-world value of LLMs was understood, scaling in LLMs became a high-priority research topic. Hoffmann et al. (2022) conducted a precise analysis into the trade-off of model and dataset size, finding they should be increased in equal proportions. This conflicted with Kaplan et al.'s suggestion that model size should be prioritized – an incorrect conclusion that Pearce and Song (2024) showed largely arose from counting only non-embedding parameters.

Many other aspects of LLM scaling analyses are beginning to be refined. Su et al. (2024) revisited the methodology used to find scaling coefficients. Hägele et al. (2024) found that multiple independent cosine schedules could be reproduced more efficiently through a constant learning rate with multiple short decays, or stochastic weight averaging. Pearce and Song (2024) & Porian et al. (2024) found that well-tuned constant learning rates were sufficient to recover certain coefficients. Bi et al. (2024) study the effect of various hyperparameters on scaling. Muennighoff et al. (2024) looked at repeated epochs, finding up to four epochs produce negligible departures from the infinite data regime. Sardana and Frankle (2023) factored in inference compute to the definition of what is compute-optimal. Isik et al. (2024) study the link between pre-training loss and downstream performance. A further line of research aims to explain *why* power laws are such a good descriptor of empirical deep learning (Hutter, 2021; Maloney et al., 2022; Bahri et al., 2024).

**Scaling in embodied AI.** Compared to LLMs, our understanding of scale in embodied settings is less advanced. Early successes in competitive games showed that running reinforcement learning (RL) at scale could surpass human performance, e.g. (Silver et al., 2017; Berner et al., 2019). In self-play RL, power laws were observed between certain quantities by Neumann and Gros (2022). Meanwhile, Hilton et al. (2023) noted that, in general, reward signals do not follow power laws, and defined a transformation of reward (intrinsic performance) that created self-consistent scaling laws.

Inspired by the effectiveness of scaling in LLMs, embodied AI research has recently begun to explore the effectiveness of generative pre-training objectives on offline datasets, when executed at scale. This includes behavior cloning objectives in video games (Baker et al., 2022; Raad et al., 2024), robotics (Brohan et al., 2022; 2023; Padalkar et al., 2023; Bousmalis et al., 2023), or multiple domains (Reed et al., 2022), as well as world modeling objectives (Hu et al., 2023; Yang et al., 2023; Bruce et al., 2024). In these studies, the benefit of scale is generally shown through increasing model size on a specific downstream task of interest (e.g. measured by completion rate) – an aggregated survey is provided by Sartor and Thompson (2024).

Whilst such studies provide strong evidence that scale can be effective in embodied domains, the complexity introduced by downstream task evaluation makes quantifying the effects, and asking more nuanced questions very challenging (e.g. how to trade-off model and dataset size). To address this, we argue that a better starting point for studying scale in these generative pre-training approaches, is to study the effect of scale on pre-training loss. Whilst downstream task performance may be of ultimate interest, LLM research evidences the utility of focusing on this clean intermediate signal, which is more straightforward to analyze. Few prior works have taken such an approach. In Appendix A we contrast our BC efforts with Tuyls et al. (2023) who provide a valuable initial study on the relationship between compute and pre-training loss for BC. We also discuss scaling analyses of image and video modeling, which are related to world modeling.

---

[1]For two variables $x$ & $y$, the power law $y = ax^b$ is invariant to scaling $x$ by a constant $c$.
Formally: $a(cx)^b = c^b ax^b \implies c^b y = c^b ax^b \implies y = ax^b$.

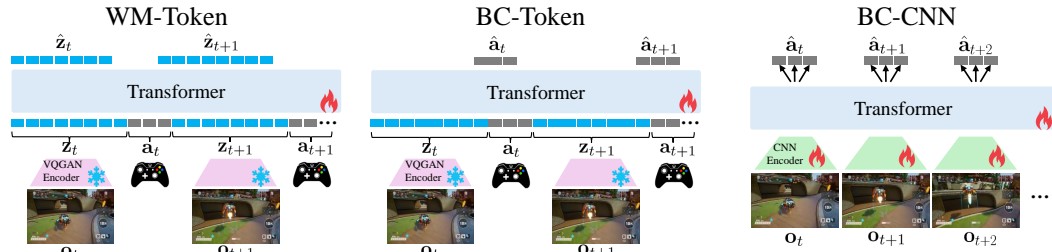

Figure 2: The World Modelling (WM) and Behavior Cloning (BC) tasks & architecture combinations considered in this work. The fire symbol signifies trainable components, the ice symbol signifies frozen pre-trained components.

## 3 METHODOLOGY

This section provides details for our main experiments. We describe the pre-training tasks, architectures, and datasets considered. We also detail the methodology used in the scaling law analyses.

### 3.1 TASKS

We consider trajectories constructed as sequences of alternating observations $\mathbf{o}_t$ and actions $\mathbf{a}_t$ for timestep $t \in \mathbb{N}$. In this work, observations are always images, $\mathbf{o}_t \in \mathbb{R}^{3 \times w \times h}$ and any continuous actions are discretized during preprocessing leaving, $\mathbf{a}_t \in \{0, 1\}^{d_a}$.

Given this data format, we consider two tasks. *World modeling* (WM) (Ha and Schmidhuber, 2018) predicts future observations from previous observations and actions. This allows an agent to explicitly understand how its environment works, which can be used for planning, or dyna-style RL (Sutton, 2018). *Behavior cloning* (BC) predicts the future actions that the dataset's demonstrators take (Bakker et al., 1996). This creates a policy that can be directly used to act in the environment, either as-is or following further fine-tuning. Concretely, these two tasks require modeling the following quantities,

$$\text{World modeling:} \quad P(\mathbf{o}_{t+1}|\mathbf{o}_t \dots \mathbf{o}_{t-k}, \mathbf{a}_t \dots \mathbf{a}_{t-k}), \tag{1}$$

$$\text{Behavior cloning:} \quad P(\mathbf{a}_t|\mathbf{o}_t \dots \mathbf{o}_{t-k}, \mathbf{a}_{t-1} \dots \mathbf{a}_{t-k-1}). \tag{2}$$

This work focuses on *generative* pre-training aiming to model this full conditional probability distribution. We leave a study of scaling laws for alternative objectives, e.g., explicitly targeting representation learning (Nair et al., 2022) or reward-centric models (Hafner et al., 2020), to future work.

### 3.2 ARCHITECTURES

All experiments revolve around GPT-2 style causal transformers (Radford et al., 2019) as the core of the model. However we consider two different methods for inputting image observations, summarized in Figure 2. Section 3.4 details how we measure the model size of each.

**Tokenized architecture.** Our first architecture tokenizes each image observation into multiple discrete tokens. This is done with a frozen VQGAN encoder $\text{Enc}_\theta(\mathbf{o}_t) \rightarrow \mathbf{z}_t$, where $\mathbf{z}_t \in \{1, 2, ..., V_o\}^{d_z}$, for vocabulary size $V_o$ and latent dimension $d_z$. Discretized actions are mapped to a non-overlapping vocabulary. Following tokenization, training sequences take the form,

$$[z_t^1, z_t^2, ..., z_t^{d_z}, a_t^1, a_t^2, ..., a_t^{d_a}, z_{t+1}^1, z_{t+1}^2, ...z_{t+1}^{d_z}, a_{t+1}^1, a_{t+1}^2, ..., a_{t+1}^{d_a}, ...], \tag{3}$$

where each item of the sequence is an integer within our vocabulary. A transformer is then trained to maximize the likelihood of either the latent image tokens (world modeling), or action tokens (BC).

This tokenized architecture is widely used both in world modeling (Micheli et al., 2022) and BC tasks (Bousmalis et al., 2023). Gato (Reed et al., 2022) used a similar design but with continuous patches rather than discrete tokens. Our implementation tests both a 'small' (28M parameters, $d_z = 256$) and 'large' (150M parameters, $d_z = 540$) VQGAN – further details in Appendix B.

**CNN architecture.** Our second architecture differs in two ways. 1) Each image observation is input into the transformer as a single continuous embedding, extracted from a small trainable convolutional neural network (CNN). 2) Action dimensions are predicted independently (rather than in series), assuming $P(\mathbf{a}_t | \dots) \approx \prod_{i=1}^{d_a} P(a_t^i | \dots)$. A single forward pass of the transformer is needed per action prediction.

This produces an architecture similar to Baker et al. (2022) (VPT additionally used a transformer-XL and a refined hierarchical action space). Our implementation uses an Impala-style (Espeholt et al., 2018) CNN with 0.6M parameters for embedding image observations.

### 3.3 DATASETS

This paper focuses on the effect of scaling on the pre-training loss over an offline dataset. To study this cleanly, datasets must meet two criteria.

1. **Dataset size.** Repeated training on the same data alters the effect of scaling. Therefore, datasets should be large enough that all model sizes use a low number of training epochs.
2. **Dataset diversity.** Both the behavior and environment must contain enough richness and variety that pre-training loss does not saturate across the model sizes tested.

Many existing benchmark datasets fail to fulfill these criteria – if not due to limited size, then because behavior is generated from a pre-trained fixed policy, or the environment is too simple.

Our work instead focuses on a dataset of human behavior collected in a video game named *Bleeding Edge*. This is a fast-paced 4 vs 4 multiplayer game, with a range of characters, abilities and maps. Game play is highly complex due to the cooperative and competitive dynamics. Success requires selecting high-level strategies (e.g. choosing which map regions to fight for), as well as fine-grained reactive control during combat. Figure 9 shows example sequences from our dataset.

Supported by the game's developer *Ninja Theory*, we compiled a dataset of 8.6 years of anonymized game play, containing both image observations and controller actions. We refer to this as the *7 map dataset*. We also use a subset of this for some experiments, of around 1.1 years from a single map, which we name the *Sky Garden dataset*. Appendix B.3 provides further technical details.

### 3.4 SCALING ANALYSIS METHODOLOGY

We are interested in studying the relationship between several quantities defined below.

- Model size $N$, the *total* number of trainable parameters (ignoring VQGAN parameters for WM-Token & BC-Token, but including the fixed-size CNN for BC-CNN). Embedding parameters are included in the count following Pearce and Song (2024).
- Dataset size $D$, the total number of *inputs* the transformer sees during training. For WM-Token and BC-Token this is $d_z + d_a$ per observation & action pair, and for BC-CNN this is one per observation & action pair.
- Compute $C$, the number of floating point operations (FLOPs) used during training. The common approximation of $C = 6ND$ (Kaplan et al., 2020) is used.
- Loss $L$, the standard classification cross-entropy loss (all targets are discretized). We assume training loss is an accurate proxy for test loss (Appendix B.3.1 analyzes further).

More specifically, we are interested in 'compute-optimal' versions of each quantity. For loss, this is defined as the minimal loss possible for a given FLOPs budget,

$$L_{\text{optimal}}(C) = \min_{\text{s.t. } C=6ND} L(N, D), \tag{4}$$

where $L(N, D)$ is the empirical loss achieved with an $N$ parameter model trained on $D$ tokens. We further define optimal model and dataset sizes as the configuration that produce this minimal loss given a FLOPs budget,

$$N_{\text{optimal}}(C), D_{\text{optimal}}(C) = \underset{N, D \text{ s.t. } C=6ND}{\text{argmin}} L(N, D). \tag{5}$$

Table 1: Summary of fitted scaling coefficients for our main experiments. Note that we favor the Frontier fit when available, and only use the Parametric fit for BC-Token-540 (see Section 3.4).

| | Frontier fit | | Parametric fit | |
|---|---|---|---|---|
| **Experiment** | $N_{\textbf{optimal}} \propto C^a$ | $D_{\textbf{optimal}} \propto C^b$ | $N_{\textbf{optimal}} \propto C^a$ | $D_{\textbf{optimal}} \propto C^b$ |
| WM-Token-256 | 0.49 | 0.51 | 0.52 | 0.48 |
| WM-Token-540 | 0.62 | 0.37 | 0.78 | 0.22 |
| BC-Token-540 | N/A | N/A | 0.32 | 0.68 |
| BC-CNN | 0.66 | 0.34 | 0.47 | 0.53 |

**Scaling analysis.** The heart of scaling law analysis is fitting power law relationships predicting these compute-optimal quantities. For predicting optimal model and dataset size, we use,

$$\hat{N}_{\text{optimal}}(C) = a_0 C^a \qquad \hat{D}_{\text{optimal}}(C) = b_0 C^b,  \tag{6}$$

with fitted constants $a_0, a, b_0, b$. [2] We consider two methods to fit these relationships, introduced by Hoffmann et al. (2022). Their Method 1, which we term *Frontier fit*, classifies efficient models as those falling on the *efficient frontier* (see Figure 1). Coefficients can then be estimated straightforwardly through a line of best fit on a plot of FLOPs vs parameters or data for these efficient models.

Frontier fit is our preferred method when available – it avoids making any assumptions about the training curves, directly fitting the best models observed. However, it requires training models past the point where they are the optimal configuration (seen on a loss-FLOPs plot as overlapping curves). In some of our experiments (BC-Token and Section 5.1), this was not possible.

In these situations, we resort to Method 3 of Hoffmann et al. (2022), which we term *Parametric fit*. This fits the coefficients $\alpha, \beta, N_c, D_c, E$ to a parametric loss form,

$$\hat{L}(N, D) = \frac{N_c}{N^\alpha} + \frac{D_c}{D^\beta} + E,  \tag{7}$$

to the empirical training curves. In our implementation, we use SciPy's `curve_fit` function. We then find $a = \beta/(\alpha + \beta), b = \alpha/(\alpha + \beta)$. This makes a very strong assumption about the training curves, but allows coefficients to be estimated at a smaller compute budget.

For loss prediction we use the form recommended by Pearce and Song (2024),

$$\hat{L}_{\text{optimal}}(N, D) = c_0 C^{-c} + E.  \tag{8}$$

We again use the `curve_fit` function, fitted to models along the efficient frontier. During fitting, we set bounds on the variables: $c_0 \in [0, \infty], c \in [-1, 1], E \in [0.1, \infty]$.

**Training details.** While early scaling studies conducted sweeps over multiple cosine decays of differing lengths (Kaplan et al., 2020; Hoffmann et al., 2022), follow up work found this redundant (Pearce and Song, 2024; Hägele et al., 2024; Porian et al., 2024). We follow the approach of using a constant learning rate per model, so each requires only one training run. We aim to train models until they have passed their compute efficient FLOPs budget. We only modify the parameters of the transformer, following the configurations documented in Appendix B.

## 4 SCALING ANALYSIS IN EMBODIED AI

This section presents our main results. We begin by considering the scaling laws for the task of world modelling in Section 4.1 with two different tokenizers (turning image observations into 256 and 540 tokens for the *small* and *large* variants respectively). Section 4.2 then considers the task of BC both with *tokenized* and *CNN* architectures. Finally, Section 4.3 tests the extrapolation capability of these scaling laws for the task of world modeling.

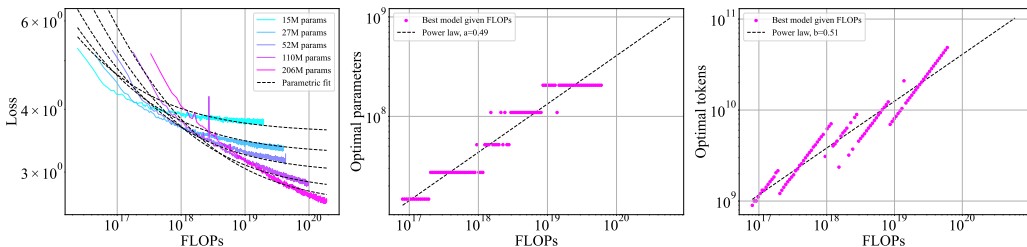

Figure 3: WM-Token scaling with $d_z = 256$ tokens per image observation. Left shows the *parametric fit*. Middle & right show the *frontier fit* estimating optimal model & dataset size respectively.

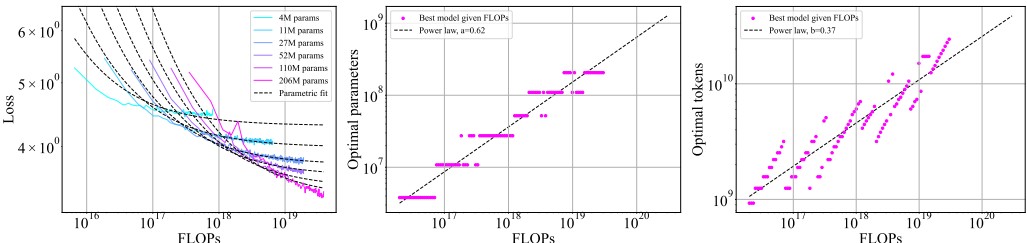

Figure 4: WM-Token scaling with $d_z = 540$ tokens per image observation. Left shows the *parametric fit*. Middle & right show the *frontier fit* estimating optimal model & dataset size respectively. Compared to the results for WM-Token-256, the power law coefficient for $N_{\text{optimal}}$ increases from 0.49 to 0.62.

## 4.1 SCALING ANALYSIS IN WORLD MODELING

Figures 3 & 4 present our results for the task of world modeling, with the scaling law coefficients summarised in Table 1. For WM-Token-256 we find that the optimal coefficients for model and dataset size are both $\approx 0.5$, e.g. one should increase both model and dataset size in the same proportions. This matches the scaling laws observed in LLMs (Hoffmann et al., 2022). Increasing the number of tokens per image to $540$ for WM-Token-540 changes the optimal trade-off between model and dataset size, skewing towards model size; $N_{\text{optimal}} = 0.62$, $D_{\text{optimal}} = 0.37$. We discuss this further in Section 5.3.

## 4.2 SCALING ANALYSIS IN BEHAVIOR CLONING

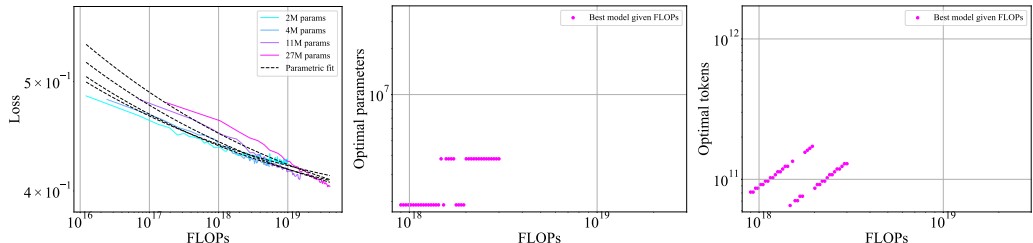

Figure 5: BC-Token scaling with $d_z = 540$ tokens per image observation. Models above 2M parameters do not saturate over the FLOPs range considered and coefficients can not be inferred using the *frontier fit* method.

We present our results on the scaling law coefficients for BC-Token in Figure 5. Despite sharing an architecture with WM-Token-540 we now observe the opposite dependence on model and dataset

---

[2]Note that by subscribing to $C = 6ND$ we find $a = 1 - b$; $N \propto C^a \implies C/D \propto C^a \implies D \propto C^{1-a}$.

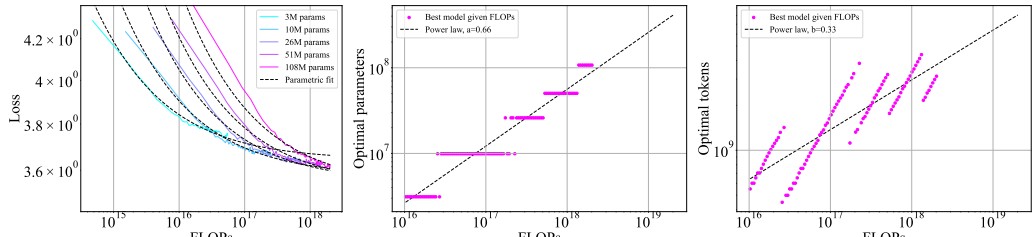

Figure 6: BC-CNN scaling. Left shows the *parametric fit*. Middle & right show the *frontier fit* estimating optimal model & dataset size respectively. Compared to the results for BC-Token the model sizes considered compute-optimal are considerably larger. The power law coefficient for $N_{\text{optimal}}$ also increases significantly from 0.32 to 0.66 skewing towards scaling model size as opposed to dataset size when scaling up compute.

sizes. The coefficients skew heavily towards dataset size; $N_{\text{optimal}} = 0.32$, $D_{\text{optimal}} = 0.68$ (compared to $N_{\text{optimal}} = 0.62$, $D_{\text{optimal}} = 0.37$ – explained in Section 5.1). Furthermore, under the same compute budget the compute-optimal model sizes are significantly smaller. For a compute budget of $10^{18}$ and $10^{19}$ FLOPs we find that model sizes of 2M and 11M are compute-optimal for BC-Token-540 compared to 27M and $110M$ for WM-Token-540. In our experiments, we observe the losses for the BC-Token models take much longer to plateau leading to less overlap between model sizes. This results in the *frontier fit* not being suitable for accurately estimating the scaling law coefficients, hence we rely on the *parametric fit* for these results.

To better understand the change in the scaling law coefficients, we now consider the BC-CNN architecture for the task of BC in Figure 6. For this architecture, we observe that the coefficients now skew towards model size (similarly to those in (Tuyls et al., 2023)), with $N_{\text{optimal}} = 0.66$, and $D_{\text{optimal}} = 0.34$. Section 5.2 provides more intuition on the differences between the WM-Token and BC-Token setups that lead to this change.

Further to studying the differences in scaling law coefficients between tasks and architectures, we also study the accuracy of extrapolation.

### 4.3 EXTRAPOLATION IN WORLD MODELING

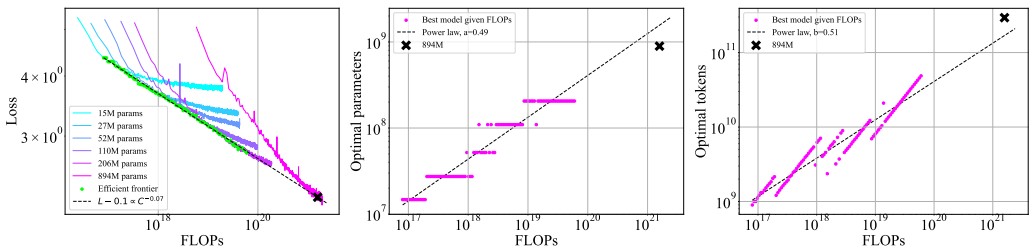

Figure 7: Testing the extrapolation capability of our derived scaling law for WM-Token-256 by training an 894M parameter model with an order of magnitude more compute than was used for the scaling law analyses. We observe good agreement between our predicted optimal loss/model size/number of training tokens (dotted lines) and our actual training run.

To test the extrapolation accuracy of our derived scaling laws, we train a $894M$ parameter WM-Token-256 model with an order of magnitude more compute than used for the scaling law analyses. Figure 7 presents both the learning curve as well as the extrapolated lines derived from the Frontier fit method. We take the point with the loss value closest to our extrapolated loss curve ($\sim 1.58 \times 10^{21}$ FLOPS), and mark it on the Frontier fit extrapolations. We observe very good agreement between that point and our compute-optimal predictions for both model and dataset size, demonstrating the accuracy of our derived scaling laws. The gap between our prediction and the actual training run suggests we could further optimize the hyperparameters (learning rate and batch

size in particular) for the 894M parameter model, which was not extensively tuned due to compute requirements.

## 5 FURTHER ANALYSIS

Section 4 made several observations about the effect of scale in the pre-training of embodied agents. This section aims to understand these results further, and provide intuition for why they occur. Specifically we target three questions.

- Q1: Why does BC-Token produce training curves that do not plateau, while WM-Token does, given an identical architecture and dataset? (Section 5.1)
- Q2: Why does moving from BC-Token to BC-CNN resolve this issue? (Section 5.2)
- Q3: Why does increasing the amount of tokens per image observation (256 to 540) lead to an increase in the optimal model size coefficient (0.49 to 0.62)? (Section 5.3)

### 5.1 Q1: BC-TOKEN VS. WM-TOKEN

The lack of saturation of BC-Token models compared to WM-Token models can be attributed to two factors. The first is a sparser loss. A single observation-action pair is discretized into $d_z + d_a$ total tokens. With the large VQGAN tokenizer, world modeling receives supervision for $d_z/(d_z + d_a) = 540/556 \approx 97\%$ tokens, while BC is supervised for $d_a/(d_z + d_a) = 16/556 \approx 3\%$ of tokens.

The second factor is the granularity of the targets. The large tokenizer creates a world modeling vocabulary size of 4096. Each vocabulary item roughly corresponds to a specific color and texture for an image patch. Many vocabulary items may only be used to model specific map regions or special abilities. Hence, the world modeling loss is very granular. On the other hand, a player can take the same action in multiple different situations – continue straight could be used to escape an enemy, chase an enemy, or navigate to a checkpoint. Hence, the supervision for BC is more vague and abstracted. We can think of this as a *super-classed* label.

To demonstrate the effect of these two factors on optimal model size coefficients, we run a set of tiny-scale experiments in language modeling. Transformers are trained on next-character prediction, on a dataset of Shakespeare text[3] using a single character for each token. Model sizes are varied from 4k parameters to 17M parameters. Context length is fixed at 16 characters/tokens.

Figure 8 (left) shows training curves over all 16 tokens, followed by a *sparse* loss where supervision is only provided from the final token (middle), and then additionally under a *super-classed* setting (right). This super-classes the final target – rather than using all 128 ASCII characters, they are randomly shuffled into one of two macro classes.

These modifications are intended to mirror the effect of moving from WM-Token to BC-Token. We compute optimal model size coefficients using the parametric fit as most models are not trained long enough for the frontier fit method. Indeed, we see that the coefficient drops from 0.63 to 0.15 with both the sparse and super-classed loss. This matches the magnitude of decrease seen in Table 1 from 0.66 to 0.32, indicating that the proposed mechanisms explain our findings.

### 5.2 Q2: BC-TOKEN VS. BC-CNN

Despite the same non-granular loss signal, why does switching architecture from BC-Token to BC-CNN makes the loss of similar model sizes plateau under a much smaller compute budget?

Consider each architecture using a transformer with 1M parameters. Observe from Figure 2 that BC-Token receives $d_z + d_a = 556$ inputs for every action $\hat{a}_t$ it predicts, while BC-CNN receives just one input for every action predicted. Hence, BC-Token uses around 556 times more compute in its action prediction ($556 \times 2 \times 1M \approx 1 \times 10^9$ FLOPs) than BC-CNN ($1 \times 2 \times 1M \approx 2 \times 10^6$ FLOPs). This means that even with the same number of parameters, BC-Token can learn a far more expressive function than BC-CNN. Hence, BC-Token requires far more tokens to match this expressivity, and training curves for a given model size plateau much later.

---

[3]Shakespeare character dataset from: `https://github.com/karpathy/nanoGPT`

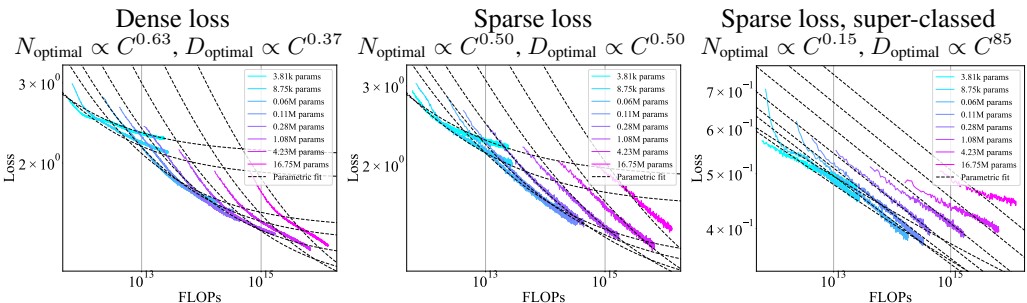

Figure 8: Training curves and parametric fit for character modeling experiments. The standard dense LLM loss has been modified to reflect properties of BC – a sparse loss (one of 16 tokens), and then additionally super-classing the targets into two classes.

### 5.3 Q3: WM-TOKEN-256 VS. WM-TOKEN-540

Finally, we seek to understand why the optimal model size coefficient increases when moving from the 256 to the 540 token VQGAN. As the number of tokens per image observation are increased, the compression rate of the tokenized representation decreases. We would expect that each individual token becomes easier to predict in this less compressed representation. This would mean a less expressive function is needed (smaller model size), but also a smaller number of examples would need to be seen (smaller dataset size). It is less clear what ratios these ingredients decrease in, and hence what effect a lower compression rate has on the optimal model size coefficient.

For further insight into how compression affects the optimal model size coefficient, we ran a small scale experiment in language modeling using two text representations; 1) ASCII character-level tokenization. (low compression) 2) GPT-2 tokenizer (high compression). We used the BookCorpus dataset (Zhu et al., 2015), and trained models past their compute-optimal point, so the *Frontier fit* method could be used for coefficient estimation.

Appendix C shows results. Under the character-level tokenizer (low compression), we find $N_{\text{optimal}} \propto C^{0.66}$. For the GPT-2 tokenizer (high compression), we find $N_{\text{optimal}} \propto C^{0.44}$. The key takeaway is that the more compressed representation has led to a lower optimal model size coefficient, which reflects what we see in the world modeling experiments.

## 6 DISCUSSION & CONCLUSION

This paper establishes a deeper understanding of scaling laws for world modeling and behavior cloning, two tasks that underpin embodied AI applications in domains such as video games and robotics. Focusing on generative pre-training of such models, we show that it is possible to recover scaling laws similar to those established in the LLM literature. Establishing such a link is key to making efficient use of available resources, and to training compute-optimal models.

Considering the task of world modeling, we find that models can be smoothly scaled following best practices and insights from the LLM literature. Surprisingly, the scaling coefficients for our WM-Token-256 architecture very closely match those established for LLMs. Comparing to our WM-Token-540 model and additional analysis, we further establish that scaling is affected by the tokenizer's compression rate.

Turning to pre-training BC policies for agents, the choice of architecture is extremely important in determining optimal scaling behavior. When using architectures with tokenized image observations, dataset size should be increased much more rapidly than model size. Meanwhile, for BC-CNN architectures, model size should be increased faster than dataset size.

**Limitations.** While we show that scaling laws can be precisely described in the infinite data regime and for appropriate architectures, future work is needed to establish scaling laws for alternative models and under varying dataset quality. In addition, we focus on loss as an intermediate quantity that can be effectively optimized in pre-training. Many additional considerations are required for effec-

tive AI models, such as downstream task performance and model inference times. How valuable scaling laws can be in providing insights relevant to those choices remains an open question.

ETHICS STATEMENT

Data for this project was provided via a partnership with *Ninja Theory*, who collected a large corpus of human gameplay data for their game *Bleeding Edge*. Data collection was covered by an End User License Agreement (EULA) and our use of the data was governed by a data sharing agreement with the game studio, and approved by our institution's IRB. This data was recorded between September 2020 and October 2022. To minimize risk to human subjects, any personally identifiable information (Xbox user ID) was removed from the data. The resulting data was cleaned to remove errors and data from bad actors.

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

The appendix is organized as follows.

- Appendix A contains an extended related work section.
- Appendix B contains details on the training of the model configurations, hyperparameters, and a description of the datasets used.
- Appendix C contains results from Section 5.3.
- Appendix D contains further results training world models on a robotics task.
- Appendix E contains further results demonstrating the link between pre-training loss and performance.

## A  EXTENDED RELATED WORK

Here we contrast more granular details of several related works.

Tuyls et al. (2023) provide (amongst other things) a valuable initial study on the relationship between compute and pre-training loss for BC. They focus on Atari and Nethack games, where the dataset is generated by a fixed pre-trained agent. They test one architecture – a single-layer LSTM network, scaling width-wise to increase parameter count. They find the model size coefficient $N_{\text{optimal}} \propto C^a$ varies between 0.58 to 0.79 for different games.

By contrast, our work focuses on transformer-based models, and we consider two variants of architecture. We also train on human behavior, rather than pre-trained agents (which could be viewed as model distillation). Additionally, we go beyond BC and also study scale in world modeling.

**Scaling laws in other domains.** Scaling laws have also been observed in auto-regressive modeling of other modalities. Most relevant to our world modeling task are those of video and images. Henighan et al. (2020) found the optimal trade off between model and dataset size to match their reported LLM coefficient ($N_{\text{optimal}} \propto C^{0.7}$) and was not affected by tokenizer. Our experiments offer different findings in the domain of world modeling on human data – we use updated methodologies to measure this trade-off (Hoffmann et al., 2022), finding it *is* affected by the tokenizer. Tian et al. (2024) show scaling laws emerge in image modeling with non raster-ordered next-token prediction.

## B  SCALING EXPERIMENTS FURTHER DETAILS

### B.1  HYPERPARAMETERS

We trained two VQGANs from scratch with reconstruction losses.

- **BE-Small.** Based on Esser et al. (2021), uses $d_z = 256$, $V_o = 4096$, $h = w = 128$, with 28M parameters, and a CNN design. It was trained on a single SkyGarden Bleeding Edge map.
- **BE-Large.** Based on Yu et al. (2022), uses $d_z = 540$, $V_o = 4096$, $h = 180$, $w = 300$, with 150M parameters, and a vision transformer design. It was trained on all seven Bleeding Edge maps.

We selected the numbers of tokens per image based on qualitative assessment of reconstructions. We found that 256 tokens per image was the minimum that still allowed a reconstruction to capture the majority of salient gameplay details. However certain details still were lacking, such as an enemy player's health bars – hence we also considered a 540 token version that provided a higher quality reconstruction.

BC-CNN details. We use $h = w = 128$. The $0.6M$ paramter CNN is similar to that used by (Baker et al., 2022), however it uses ConvNext blocks (Liu et al., 2022). The CNN produces an embedding of size $1024$ which is then put through a linear layer to obtain a vector matching the transformer's embedding dimension.

Transformer configurations are given in Table 2. We describe the parameters for the WM-Token architecture. Note that MLP layers are four times the width of embed dim. Model configurations

roughly followed the model configurations used in Table A9 of Hoffmann et al. (2022), where residual stream dimension, number of layers, and number of heads were roughly increased proportionally.

Table 2: Transformer configurations. Here $N$ is listed for the tokenized architectures. Parameter count varies slightly for BC-CNN due to inclusion of the embedding CNN and differing numbers of embedding parameters sizes.

| $N$ | Layers | Num heads | Embed dim |
|---|---|---|---|
| 2M | 3 | 3 | 180 |
| 4M | 4 | 4 | 240 |
| 11M | 6 | 6 | 360 |
| 15M | 4 | 4 | 512 |
| 27M | 8 | 8 | 512 |
| 52M | 10 | 10 | 640 |
| 110M | 15 | 12 | 768 |
| 206M | 16 | 16 | 1024 |
| 894M | 23 | 14 | 1792 |

## B.2 TRAINING DETAILS

All transformers are trained with a variant of nanoGPT (Karpathy, 2022) using PyTorch Lightning (Falcon and The PyTorch Lightning team, 2019).

This section lists key hyperparameters. Note that it was important to find optimization settings that produced the lowest possible loss for a given model size. In general larger models require smaller learning rates. Our approach first optimized the smallest model through a grid sweep, we would then sequentially run a sweep over the next largest model, starting at the smaller model's optimized learning rate. Table 3-6 provide final settings.

Table 3: Hyperparameters for WM-Token with $d_z =256$ tokens per image observation.

| $N$ | Seq len | Context length | Tokens per update | Learning rate |
|---|---|---|---|---|
| 15M | 10 | 2,720 | 522,240 | 0.0007 |
| 27M | 10 | 2,720 | 522,240 | 0.0007 |
| 52M | 10 | 2,720 | 522,240 | 0.0007 |
| 110M | 10 | 2,720 | 522,240 | 0.0007 |
| 206M | 10 | 2,720 | 522,240 | 0.00057 |
| 894M | 10 | 2,720 | 2M | 0.00028 |

Table 4: Hyperparameters for WM-Token with $d_z =540$ tokens per image observation.

| $N$ | Seq len | Context length | Tokens per update | Learning rate |
|---|---|---|---|---|
| 4M | 10 | 5,560 | 533,760 | 0.005 |
| 11M | 10 | 5,560 | 533,760 | 0.001 |
| 27M | 10 | 5,560 | 533,760 | 0.001 |
| 52M | 10 | 5,560 | 533,760 | 0.001 |
| 110M | 10 | 5,560 | 533,760 | 0.0005 |
| 206M | 10 | 5,560 | 533,760 | 0.0005 |

Table 5: Hyperparameters for BC-Token with $d_z =$ 540 tokens per image observation.

| $N$ | Seq len | Context length | Tokens per update | Learning rate |
|---|---|---|---|---|
| 2M | 10 | 5,560 | 533,760 | 0.0005 |
| 4M | 10 | 5,560 | 533,760 | 0.0005 |
| 11M | 10 | 5,560 | 533,760 | 0.0001 |
| 27M | 10 | 5,560 | 533,760 | 0.0001 |

Table 6: Hyperparameters for BC-CNN.

| $N$ | Seq len | Context length | Items per update | Learning rate |
|---|---|---|---|---|
| 2M | 10 | 10 | 2560 | 0.0003 |
| 3M | 10 | 10 | 2560 | 0.0003 |
| 10M | 10 | 10 | 2560 | 0.0003 |
| 26M | 10 | 10 | 2560 | 0.0003 |
| 51M | 10 | 10 | 2560 | 0.0003 |

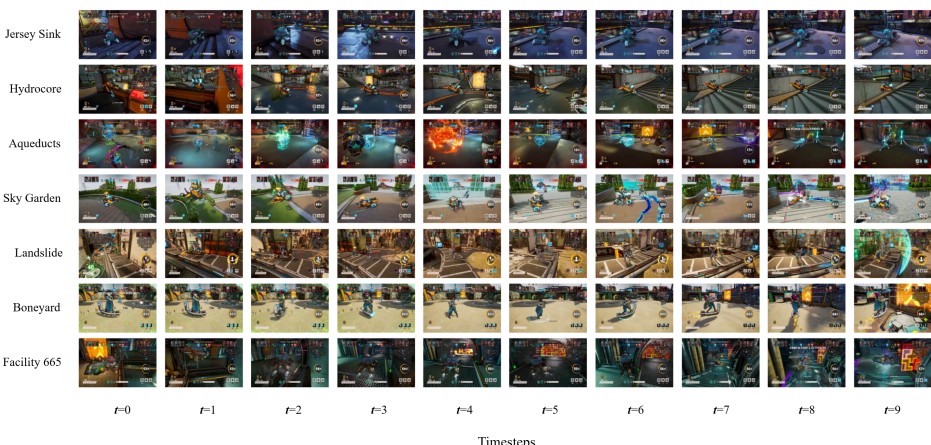

Figure 9: Example trajectories from a dataset of 8.6 years of human gameplay in the video game *Bleeding Edge* across 7 maps.

### B.3 DATASET DETAILS

Image observations were stored in MP4 format at 60fps, alongside binary files containing the associated controller actions. A time code extracted from the game was stored for each frame, to ensure actions and frames remained in sync at training time.

The *7 Maps* dataset comprised 60,986 matches, yielding 530,713 individual player trajectories (each around 9 minutes), totaling 27.89 TiB on disk. This amounted to around 8.6 years of gameplay. After downsampling to 10Hz (the frequency models are trained on), this equated to 1.63B frames. This was then divided into training / validation / test sets by dividing the matches with an 80:10:10 split.

Our filtered *Sky Garden* dataset used the same 80:10:10 split and 10Hz downsampling, but focused on just one map, yielding 71,940 individual player trajectories, or 355.5M frames (around 1.12 years of game play).

For discretizing the controller actions, while the buttons are natively discrete, we discretize the x and y values of the left and right joysticks into eleven buckets.

### B.3.1 ON THE INFINITE DATA REGIME

We wish to study scaling in the infinite data regime, where training loss is not significantly effected by models repeatedly training on the same datapoints which can lead to overfitting effects. This section calculates the number of training tokens allowed for each model family trained in this work. Viewing Figure 1 alongside these numbers confirms that models remain in the infinite data regime for all our experiments.

**WM-Token-540, BC-Token-540.** We trained on the *7 maps* dataset, with 1.63B observation-action pairs. Models used the tokenized architecture with the large VQGAN, so each observation-action pair creates $540 + 16 = 556$ transformer inputs, for a total of $1.63\text{B} \times 556 = 906\text{B}$ training tokens. Muennighoff et al. (2024) observe that tokens may be reused up to four times with negligible departure from the infinite data regime. This produces 3.6T tokens. For a 200M parameter model the compute allowed by the infinite data regime is $C = 6ND = 6 \times 200\text{M} \times 3.6\text{T} = 4.3 \times 10^{21}$ FLOPs.

**WM-Token-256.** This is trained on the *Sky Garden* dataset, with 355M observation-action pairs. Each pair is split into $256 + 16 = 272$ tokens, for 97B training tokens, or $97\text{B} \times 4 = 386\text{B}$ effective tokens. For a 200M parameter model the compute allowed by the 'infinite data regime' is $C = 6ND = 6 \times 200\text{M} \times 386\text{B} = 4.6 \times 10^{20}$ FLOPs.

**BC-CNN.** Trained on *7 maps* dataset, but now with one token per observation-action pair, this creates a possible $1.63\text{B} \times 4 = 6.52\text{B}$ effective tokens. A 50M parameter model uses $C = 6ND = 6 \times 50\text{M} \times 6.52\text{B} = 2.0 \times 10^{18}$ FLOPs.

## C FURTHER ANALYSIS DETAILS

Experimental results supporting Section 5.3.

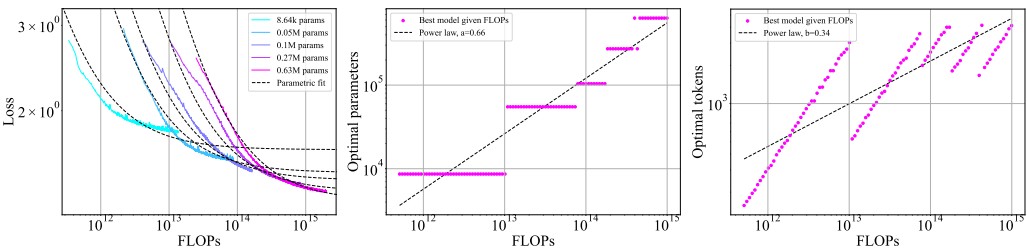

Figure 10: Relating to Section 5.3, character-level (low compression). Utilising the *frontier* fit (middle and right) we derive the power law coefficient for $N_{\text{optimal}}$ as $0.66$ and $D_{\text{optimal}}$ as $0.34$.

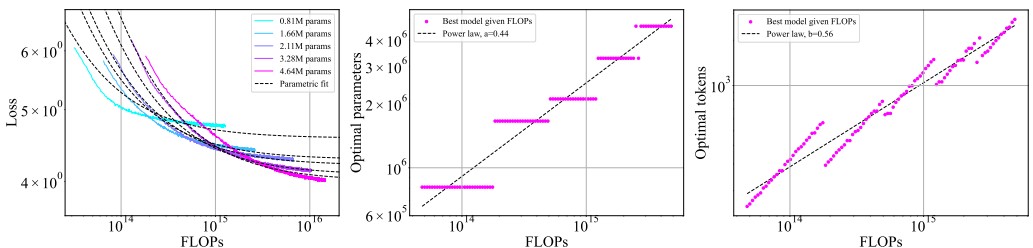

Figure 11: Relating to Section 5.3, GPT-2 tokenizer (high compression). Utilising the *frontier* fit (middle and right) we derive the power law coefficient for $N_{\text{optimal}}$ as $0.44$ and $D_{\text{optimal}}$ as $0.56$, an increase from $0.66$ in Figure 10 found when utilising a lower compression character-level tokenizer.

## D  WORLD MODELING FOR ROBOTICS

To supplement our main experiments conducted on a large dataset of human behavior in a video game, we conduct smaller scale experiments on the real-world robotics RT-1 dataset (Brohan et al., 2022) – a dataset of high-skill human demonstrations. These firstly evidence that scaling laws emerge in other environments. Furthermore, we find further support that tokenizer compression rate (number of tokens per observation) affects scaling coefficients.

We use the RT-1 dataset, resized to 128x128 pixels per image. We trained a set of five VQVAEs with $z_o \in [16, 36, 64, 100, 256]$ and $V_o = 4096$ for 40,000 updates on batches of 128. Reconstructions are visualized in Figure 12.

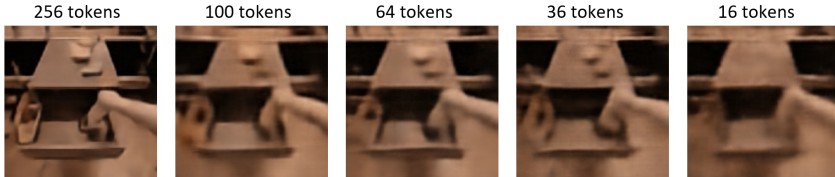

Figure 12: VQVAE reconstructions on the RT-1 dataset for differing numbers of tokens per observation, $z_o \in [16, 36, 64, 100, 256]$.

For each VQVAE, we then train a range of WM-Token model sizes $N \in [0.08M, 0.2M, 0.28M, 0.54M, 0.99M]$, and measure scaling coefficients using the frontier fit method – see Figure 14 for training curves.

First, we observe that scaling laws similar to those in our main set of experiments emerge for all VQVAEs, with predictably decreasing losses and similar optimal parameter scaling coefficients (0.56 to 0.65 in Figure 14).

Secondly, we observe that the optimal parameter scaling coefficient increases with increasing numbers of tokens per image. To verify this, we repeated three times, and in Figure 13 plot all coefficients. This supports our claim that a higher compression rate leads to a lower optimal parameter coefficient.

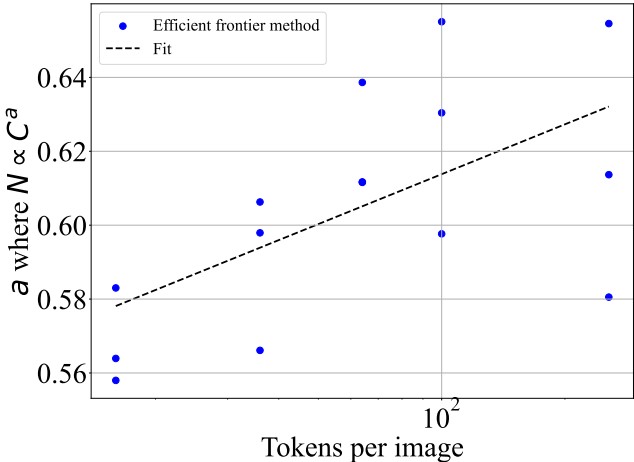

Figure 13: RT-1 experiments. Optimal parameter coefficient vs. number of tokens per observation, with three repeated runs per VQVAE.

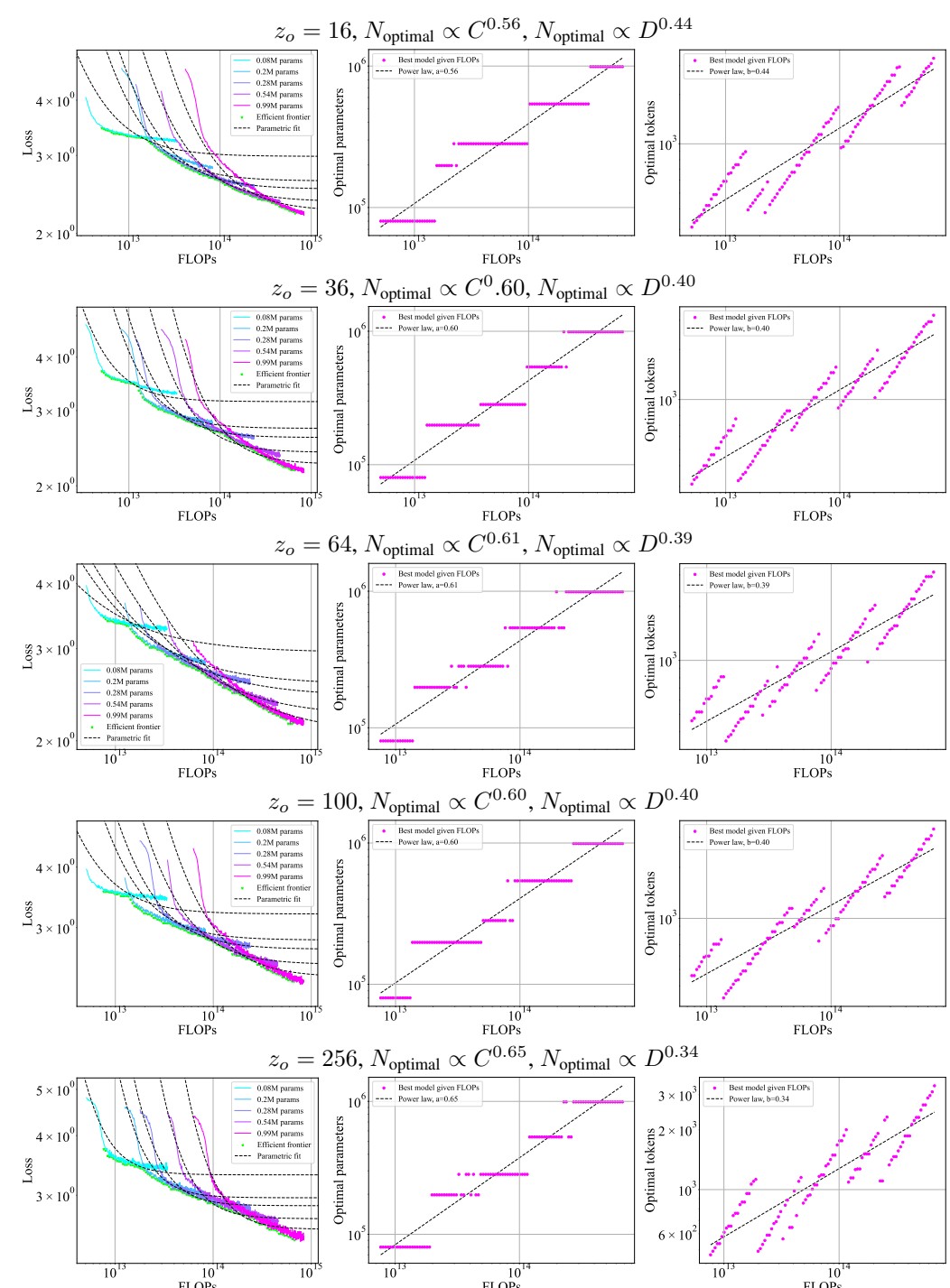

Figure 14: RT-1 experiments. Note that the optimal parameter coefficient increases with the number of tokens per observation.

.

# E  PRE-TRAINING LOSS AND PERFORMANCE

This section presents evidence for pre-training loss correlating with agent performance.

## E.1  PRE-TRAINING LOSS VS. WORLD MODELING METRICS

We use metrics commonly used to assess the quality of the world models (Yang et al., 2023), originally developed in the video generation literature. Conditioned on an initial real frame and a sequence of real actions, we compare the observations generated by the world model, with the real sequence of observations, measuring FVD, LPIPS and PSNR. Specifically, we generate 1024 videos each of 10 seconds. We perform this for various checkpoints on each size in our WM-Token-256 set of models. This allows a plot of the checkpoint pre-training loss vs video generation metric to be assessed.

Figure 15 shows results. We find correlations of 0.77, -0.57, 0.83 for LPIPS, PSNR, FVD respectively. This evidences the strong relationship between pre-training loss and world model quality.

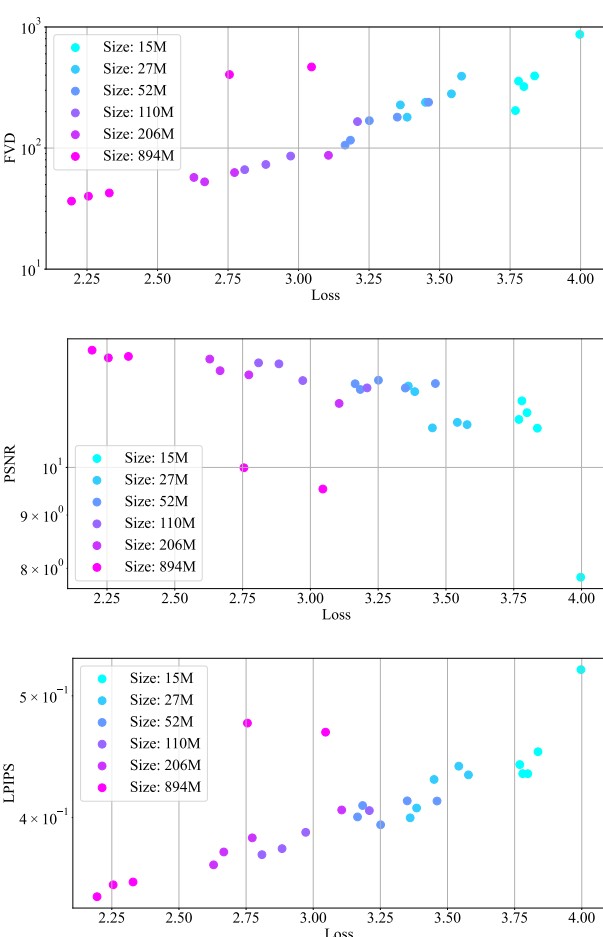

Figure 15: WM-Token-256 pre-training loss vs. video generation metrics. Pearson correlation of 0.77, -0.57, 0.83 is found for LPIPS, PSNR, FVD respectively.

## E.2  BEHAVIORAL CLONING META-ANALYSIS

We conducted a meta-analysis of Tuyls et al. (2023) to quantify the strength of relationship between pre-training loss and online return. This is possible in their set up for two reasons. 1) Tuyls et al.

use a simple environment that can be run cheaply as a gym environment, so all checkpoints can be evaluated. 2) Tuyls et al.'s data generating policy is an expert rules-based policy, so BC necessarily leads to an expert agent.

By pairing data points from Figure 5a and b of Tuyls et al. (2023), we plot pre-training loss vs. online return in Figure 16. This produces a correlation coefficient of -0.97, which provides strong evidence that in the infinite data regime, pre-training loss is a good proxy for online performance.

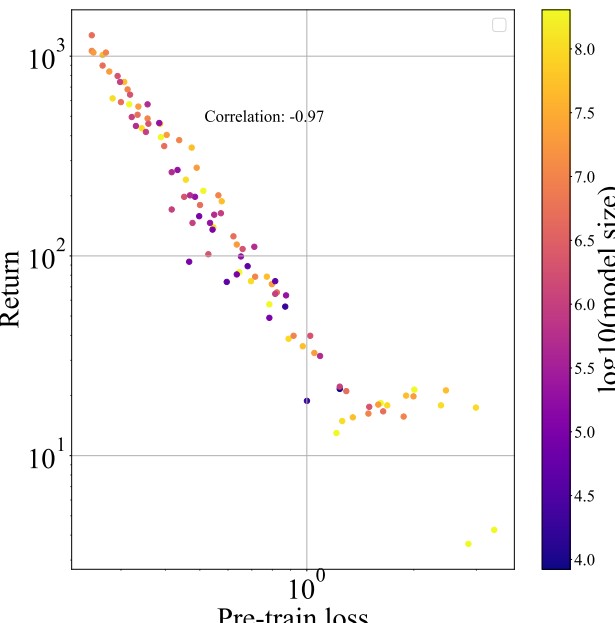

Figure 16: Meta analysis of Tuyls et al. (2023), assessing strength of the relationship between pre-training loss and online return. This came out to a Pearson correlation coefficient of -0.97.