# OpenReview forum: "Scaling Laws for Pre-training Agents and World Models"
_ICLR.cc/2025/Conference — Submitted to ICLR 2025_

### Official Review · Reviewer_Qufp · 2024-11-03

**Soundness:** 3
**Presentation:** 2
**Contribution:** 2
**Rating:** 5
**Confidence:** 3

**Summary:**

Less is understood about scaling in embodied learning, but recent works suggest that increasing model and dataset size can lead to capable agents. This paper explores this hypothesis further by delving into scaling laws applicable to pre-training embodied agents and their world models, leveraging larger datasets and enhanced computational resources to improve performance. Focusing on both world modeling and behavior cloning, the study extends the understanding of how scaling affects these models beyond simple increases in parameter count, introducing dependencies on factors like tokenization and model architecture.

**Strengths:**

#### Strengths:
1. **Empirical Validation:**
   - The paper provides evidence that scaling laws known from LLMs similarly apply to world modeling and BC, validated across some architectures and tokenization strategies.
   - Thoroughness of many results.
   - The systematic experimentation using different model sizes and computational scales support the credibility of the findings.

2. **Take on Tokenization and Architecture:**
   -  Impact of tokenization granularity on optimal model scaling is particularly novel, offering insights into how token compression influences model efficiency and effectiveness.
   - The comparison between tokenized and CNN based approaches in BC provides nice distinctions in how these architectures manage to scale.

3. **Methodology and Analysis:**
   - The paper details the methodologies employed, from the scaling analysis approach to breakdown of the computational and parameter scaling. This thoroughness can ensure clarity and reproducibility of the results.
   - The use of frontier fit and parametric fit methods to analyze scaling laws provides a usable framework for understanding the optimal trade-offs between model size and computational expense.

**Weaknesses:**

#### Areas for Improvement:
1. **Hyperparameter Selection and Rationale:**
   - While the paper provides a framework for understanding scaling laws, a more granular justification of the choices behind specific hyperparameters, particularly in the synthetic data generation and scaling analyses, would enhance its contribution. Specifically, selection of number of tokens per image in the tokenization process and the learning rates used across different model scales seem crucial to the study’s outcomes but lack a detailed rationale. Providing insights into how these parameters were optimized, may be through sensitivity analyses or referencing empirical benchmarks, could help.
   - Insights into the selection process for training configurations across different scales could further enhance the paper’s utility to practitioners.

2. **Broader Applicability and Generalization:**
   - The study provides insights into scaling laws within controlled experimental setups. However, to enhance the relevance and applicability of these findings to real-world applications, it would be beneficial to extend the analysis beyond the current datasets. Specifically, exploring how these scaling laws hold across datasets with higher environmental variability and some degree of realism or some evidence of extensibility.
   - Extending the analysis to include more diverse datasets or environmental complexities could help validate the laws observed.

3. **Limitations and Future Work:**
   - The paper acknowledges limitations in the scope of architecture and tokenization diversity.

4. **Less emphasis on embodied AI:**
   - my main criticism concerns the paper's lack of a cohesive narrative explaining the relevance of its findings to embodied intelligence. It is currently less sound in terms of the necessity of the study, or why BC or world modelling techniques need this study particularly. is it because they are the most promising approaches.. or are they the most compelling representative candidates of model-free and model-based approaches or where are we heading with this analysis? Perhaps this can be easily fixed by revising some of the scaling law papers for sing-agent or multi-agent papers cited in the paper. Lastly, might be missing some relevant references and citations in the area, some are highlighted below and good to cite.

[1] GenRL: Multimodal-foundation world models for generalization in embodied agents, Mazzaglia et al

[2] Daydreamer: World models for physical robot learning, Wu et al

[3] ArCHer: Training Language Model Agents via Hierarchical Multi-Turn RL Zhou et al.

[4]Choreographer: Learning and Adapting Skills in Imagination, mazzaglia et al

**Questions:**

This paper makes promising contributions to the understanding of scaling laws in the context of pre-training for embodied AI, with clear strengths in empirical validation and methodological rigor. Addressing the identified gaps in hyperparameter transparency and broadening the scope of environments and architectures considered could further enhance its impact. Moreover, clarifying how this research fits into the broader context of embodied AI or RL, specifically the types of tasks and applications targeted and the importance of scaling laws for these areas would make the findings more relevant and engaging. Currently, the title is a bit ambiguous or too broad to understand the scope. I am happy to increase the score post-author response.

---

> ### Author Response · Authors · 2024-11-27
> **Response to Reviewer Qufp**
>
> Thank you for your review. We are pleased to have been able to communicate the strengths of our work. Of your three main feedback points, we believe we have been able to concretely provide evidence for two (further detail on hyperparameters, and experiments in a new environment), and we elaborate on our view on the third. Please let us know if you deem this sufficient to justify an improvement to your score?
>
> ## Hyperparameter Selection and Rationale
>
> We provide further detail about hyperparameters below. This has been incorporated into the new paper version.
>
> __Tokens per image.__ For the VQGANs, we selected two numbers of tokens per image based on qualitative assessment of reconstructions. We found that 256 tokens per image was the minimum that still allowed a reconstruction to capture the majority of salient gameplay details. However certain details still were lacking, such as an enemy player's health bars -- hence we also considered a 540 token version that provided a higher quality reconstruction. Our newly added robotic experiments explore an even wider range; 16, 36, 64, 100, 256.
>
> __Learning rates.__ It was important to find optimization settings that produced the lowest possible loss for a given model size. In general larger models require smaller learning rates. Our approach first optimized the smallest model through a grid sweep, we would then sequentially run a sweep over the next largest model, starting at the smaller model's optimized learning rate. Table 3-6 provide final settings.
>
> __Model configs.__ Model configurations roughly followed the model configurations used in Table A9 of [1], where residual stream dimension, number of layers, and number of heads are roughly increased proportionally. Table 2 provides details.
>
> __Synthetic data generation.__ Note that while our environment is a simulated video game world, the dataset is not simulated -- all trajectories are from real humans playing the game. Please let us know if you'd like to see any specific statistic or detail included, beyond those in Appendix B.3.
>
>
> ## Broader Applicability and Generalization
>
> Please see our global response 1.2 for detail on our additional experiments on a new robotics dataset, which demonstrates the same scaling patterns seen in our main set of results.
>
> ## Implications for embodied AI and RL
>
> The embodied AI community has recently seen a trend towards pre-training large models on large amounts of generic behavioral data, using simple generative objectives of BC [4, 5, 6] and world modeling [2, 3]. This training can either be done in isolation, or can be followed by a fine-tuning phase doing BC on expert-only data or alternatively rewards-driven RL [6].
>
> It is our belief that a more scientific understanding of this pre-training process is of high value to such large-scale efforts. For example, allowing optimal sizing of models for a given compute budget, and informing the impact of architectural choices on scaling behavior.
>
> More generally, providing strong evidence that models trained on these embodied AI objectives predictably improve with more resources is a simple, but vitally important takeaway for the entire community.
>
>
> ## References
>
> [1] Training Compute-Optimal Large Language Models
> [2] Genie: Generative Interactive Environments
> [3] Learning Interactive Real-World Simulators
> [4] A Generalist Agent
> [5] RT-2: Vision-Language-Action Models Transfer Web Knowledge to Robotic Control
> [6] Video PreTraining (VPT): Learning to Act by Watching Unlabeled Online Videos

---

### Official Review · Reviewer_QYZ4 · 2024-11-03

**Soundness:** 3
**Presentation:** 3
**Contribution:** 1
**Rating:** 5
**Confidence:** 4

**Summary:**

This paper studies scaling laws in the context of agents (specifically in a video game) to study whether and how these laws exist in this setting. The paper looks at how dataset and model size can be optimally scaled to optimize loss for BC and world modeling (next state prediction). The authors find a power law relationship and show how things such as tokenization can effect the optimal scaling parameters.

**Strengths:**

The paper is well presented and generally clear.

The analyses are well explained and clear to read.

There are a lot of interesting analyses done, the authors really tried a lot of things here and have a lot of results to pick through and analyze.

**Weaknesses:**

The premise of what this paper is doing is kind of questionable. Specifically, the thing that is being measured is training (will get to that) loss on a BC or state prediction objective, rather than a direct measure of downstream performance. As the authors point out in their related work, most other papers such as Hu et al 2023 do measure downstream performance. The author's justification for this choice then is that this adds complexity and it might be difficulty to ask "more nuanced questions" such as how to trade-off model and dataset size. But this isn't really further justified any further. Why couldn't we do a study looking at that tradeoff on the downstream task. More importantly, this paper draws conclusions based on the training loss for a *proxy* to what researchers actually care about (how well the final agent actually does at test time), and given the choice between looking at this paper's measurements and one that evaluates this, I don't know why you would prefer this. There isn't really even much analysis about why this is a good proxy metric (like some kind of finding that things with lower loss in this paper do in fact do better on the final task, even if the curves are not as smooth)

Not only this, but the paper isn't even looking at the Test loss in these analyses, it's looking at train loss. From S3.4 "We assume training loss is an accurate proxy for test loss. (Appendix B.3.1 analyzes further)." I find this assumption both baffling and completely unjustified in the text of the paper. The cited appendix merely looks at the size of the data they trained on and says "The compute allowed by the infinite data regime is 6ND" (and so they say, they are above that. There isn't a citation for this anywhere, and I am not familiar enough with the literature to tell if this is a thing other works have done. The original scaling law paper as far as I can tell does not do this. But I still do not understand the decision. If you have this large a dataset, surely you have enough data to have a held-out set? Again, scaling laws are not something I was intimately familiar with, so if there is a good justification for this, someone please let me know. But this paper does not justify it and on it's face "assume train and test are the same" feels like an ML sin.

The results are also only validated on a single environment / datasets. This seems like it really is limiting the potential applicability of the paper. What if the conclusions or scaling parameters found in this paper are idiosyncratic to this particular game?

I say this somewhat advisedly, but I am not actually sure the models studied here are large enough to fully justify all the conclusions. Most of the experiments use 206M parameters with many having even smaller (e.g. 27M for Figure 5). The only experiment with a larger size is Figure 7 at 894M. This isn't my biggest criticism, and I know compute is quite expensive, but it feels like this is a major missing thing here.

**Questions:**

See above

What is this paper doing or concluding that is above prior work?

Why should I trust this proxy loss (train versus test and not final performance) over other works which directly measure the variable of interest?

---

> ### Author Response · Authors · 2024-11-27
> **Response to Reviewer QYZ4**
>
> Thank you for your feedback, which evidently was written with a good deal of care. We have taken great efforts to accommodate your three main feedback points.
> 1) We were able to straightforwardly clarify our simple miscommunication of the train loss vs validation loss issue.
> 2) We have responded at length to your query about why pre-training loss is a useful thing to focus a scaling study on -- a point we agree the original paper version did not justify in sufficient detail.
> 3) We have incorporated new experiments on world modeling on a new robotics dataset.
>
> We hope you might feel this is sufficient to consider an uplift in your score.
>
> ## Focus on scaling pre-training loss rather than downstream task performance
>
> Please see our global response 1.1.
>
> ## Train loss rather than test loss
>
> Apologies for not making this more clear -- this indeed is common practice in scaling law analyses, for example [3] explain `the smoothed training loss is an unbiased estimate of the test loss, as we are in the infinite data regime (the number of training tokens is less than the number of tokens in the entire corpus)'. Since each training batch has never been seen previously, the loss recorded on it is equivalent to the loss of some held out test dataset.
>
> To check that we are not significantly repeating data usage in our training runs, we conduct an analysis in Appendix B.3.1, which estimates the maximum FLOPs for a given model size and dataset that would remain in the infinite data regime.
>
> ## Lack of task diversity
>
> Please see our global response 1.2.
>
> ## Models on the smaller side
>
> We agree that our models are smaller relative to those studied in the language domain. We point out that smaller model sizes are common in world modeling and BC, and our largest world models are only around one order of magnitude less than recent state-of-the-art world models -- Genie 2.3B [1], Unisim 5.6B [2].
>
> ## Contributions over prior work
>
> Please see our global response 1.3.
>
> ## References
>
> [1] Genie: Generative Interactive Environments
> [2] Learning Interactive Real-World Simulators
> [3] Training Compute-Optimal Large Language Models

---

> ### Comment · Reviewer_QYZ4 · 2024-12-02
>
> Thank you for the detailed response.
>
> The clarification on training loss is really helpful. And the meta-analysis on the pretraining loss is somewhat helpful as is the addition of the robotics data. I’ve increased my score to reflect these clarifications/improvements.
>
> I still remain wary of relying so much on the proxy metric here and even with the robotics task, it’s still hard for me to really see now two domains as sufficient to make general statements about scaling laws in environments. I think the paper would be strengthened by evaluating on a suite of tasks (eg on offline RL datasets which incorporate many different environments).

---

> ### Author Response · Authors · 2024-12-03
>
> Thank you for taking the time to read our rebuttal and update your score. We are pleased to have been able to address some concerns.

---

### Official Review · Reviewer_sTyb · 2024-11-07

**Soundness:** 3
**Presentation:** 2
**Contribution:** 2
**Rating:** 5
**Confidence:** 3

**Summary:**

This paper explores the scaling law within an embodied agent setting, analyzing two tasks: world modeling and behavior cloning. The results indicate that a scaling law exists for both tasks when additional compute resources are provided.

**Strengths:**

- The paper focuses on an important problem that will have a large impact on the community.

**Weaknesses:**

Although it is a good attempt at analyzing the scaling law, the analysis lacks comprehensiveness.
 - The relationship between loss and dataset size, which is mentioned in the introduction, is not presented. For example, in line 53, the authors state, “The optimal trade-off between model and dataset size in world modeling is influenced by the tokenizer’s compression rate (number of tokens per observation) (Section 4.1, Figure 1a & b).” However, Figures 1a and 1b primarily illustrate the effects of computation with different tokenizers and model sizes, without clarifying the influence of dataset size on the results.
 - The experiments utilize a frozen VQGAN visual encoder, which implies that the final claims regarding tokenizers are inherently limited by the performance of the frozen encoder.
 - The architectural analysis includes a set of BC-CNN experiments under the BC loss, but it is unclear why the CNN experiment was not conducted within the world modeling tasks.

**Questions:**

Please refer to the weakness section.

---

> ### Author Response · Authors · 2024-11-27
> **Response to Reviewer sTyb**
>
> Thanks for your feedback. We respond to your queries below, in particular pointing out one issue is simple matter of miscommunication. We hope this might be sufficient to consider upgrading your score.
>
> ## Relationship between loss and dataset size not described
>
> We apologize for not making this clear. It's true that in the paper text we focus on presenting the relationship between optimal parameters $N$ and compute $C$, $N \propto C^a$. However, since we use the approximation $C=6ND$ throughout the paper, the relationship between optimal dataset size $D$ and compute is implied -- $N \propto C^a \to C/D \propto C^a \to D \propto C^{1−a}$. So for example in Figure 1, when we wrote $N_\text{optimal} \propto C^{0.49}$, this implies, $D_\text{optimal} \propto C^{0.51}$. To avoid future confusion, we have added the dataset size relationships to all plots, and added a further explanation to the paper. Thank you for drawing our attention to this.
>
> ## Frozen VQGAN visual encoder
>
> It's true that the quality of the final policies and world models are limited by the amount of information in the representation from the VQGAN. At no point, do we make comparisons between the quality of policy or world model using different encoders -- the pre-training losses are only compared between models using the same encoders.
>
> ## Lack of CNN encoder with WM task
>
> We chose to use the most popular generative modeling approaches to world modeling and BC. It is possible to consider a world modeling architecture with a single continuous embedding as input, but it is not clear what the reconstruction target and loss be -- use VQGAN tokens, or an MSE error in pixel space? To our knowledge, this is not a common world modeling architecture, so we have excluded it from our tested architectures.
> Please share a paper title or link to open source code to specific examples you would like to see included?

---

### Official Review · Reviewer_7k6m · 2024-11-10

**Soundness:** 2
**Presentation:** 3
**Contribution:** 2
**Rating:** 3
**Confidence:** 4

**Summary:**

## Summary
This paper investigates the impact of model size, dataset size, and compute on embodied AI tasks like behavior cloning (BC) and world modeling (WM). Using transformers with tokenized inputs or CNN embeddings, the authors explore scaling behavior using 8.6 years of gameplay data from *Bleeding Edge*. They find that scaling laws—typically observed in language modeling—also apply to embodied AI tasks, with different trade-offs influenced by task, architecture, and tokenization. For instance, in the WM task with 540-token representations, optimal model size scales as $N(optimal) ∝ C^{0.62}$, whereas BC tasks see optimal model size scaling as $ N(optimal) ∝ C^{0.32} $ with the same tokenized setup.

## Quality & Clarity
The paper is written clearly, with effective visualizations such as Figures 3 and 4 showing the loss curves and power law fits for scaling laws across various model sizes. The methodology in Table 1 provides fitted coefficients across model-dataset configurations. In Section 5, several design choices, such as the comparison of architectural choices for BC-Token versus BC-CNN, are explored, and explanations are provided for why specific scaling trends arise in each architecture.

## Recommendation
The paper contributes to the understanding of scaling laws for embodied AI, particularly with evidence of model-dataset trade-offs for BC and world modeling tasks on a single dataset. However, the contributions are limited by their similarity to previous work (e.g., Tuyls et al.) and the relatively narrow focus on architectural adjustments rather than task diversity or downstream metrics. I currently recommend rejection but would be willing to change my stance if the aforementioned weaknesses are addressed.

**Strengths:**

## Strengths
1. **Visual Encoder Architecture Exploration:** The study explores scaling laws across various visual encoding architectures and pre-training tasks including BC and WM, showing that model scaling behavior in embodied AI resembles language modeling. For example, WM tasks with 256-token representations require scaling model and dataset sizes equally $N(optimal) ∝ C^{0.49}$ and $ D(optimal) ∝ C^{0.51} $.
2. **Tokenizer compression:** The authors find that tokenizer compression rates significantly affect scaling laws. For instance, increasing token compression from 256 to 540 tokens per image in the WM task shifts optimal model scaling to $N(optimal) ∝ C^{0·62}$, favoring larger model sizes.

**Weaknesses:**

## Weaknesses
1. **Lack of real-world data:** The scaling insights are based on a single simulation task. Previous studies have demonstrated that simulation performance often does not correlate well with real-world robot performance [1]. The analysis would benefit from validation on openly available real-world datasets such as OpenX[2] and DROID[3].
2. **Downstream task performance:** The paper focuses exclusively on scaling pre-training loss without examining generalization or performance on downstream tasks, which is crucial for embodied AI applications. Including evaluations on even a select set of downstream tasks would significantly strengthen the practical relevance of the findings.

## References
1. An Unbiased Look at Datasets for Visuo-Motor Pre-Training: Dasari et al.
2. Open X-Embodiment: Robotic learning datasets and RT-X models: OXE Collaboration
3. DROID: A Large-Scale In-the-Wild Robot Manipulation Dataset: Khazatsky et al.

**Questions:**

## Questions
1. Would incorporating large-scale real-world robotics data help validate the paper's scaling law claims beyond simulation environments? The current findings, while valuable, are limited to the simulation world.
2. Could the authors demonstrate the practical value of lower pre-training losses by evaluating a subset of pre-trained models on downstream tasks? This would help establish a clear correlation between pre-training performance and downstream task effectiveness.

---

> ### Author Response · Authors · 2024-11-27
> **Reponse to Reviewer 7k6m**
>
> Thank you for taking the time to review our paper. We respond to your queries below. Please let us know whether you feel this meets the criteria you outlined for increasing your score -- in particular our meta analysis showing strong correlation between pre-training and downstream metrics, and new experiments in robotics using a subset of the OpenX dataset mentioned in your review.
>
> ## Lack of real world data
>
> Regarding task diversity, please see our global response 1.2.
>
> Our new experiments on world modeling for real-world robotics show pre-training loss decreases in the same way as observed in our video game dataset. Regarding [1], we note that they show online performance in a simulated environment only roughly corresponds ($R^2$=0.34) to online performance in a real environment. Whilst our paper indeed focuses on a simulated environment, we focus on pre-training loss rather than online performance, so it is unclear how relevant the findings are.
>
> ## Focus on scaling pre-training loss rather than downstream task performance
>
> Please see our global response 1.1.
>
> ## Contributions over prior work
>
> Please see our global response 1.3.
>
> ## References
>
> [1] An Unbiased Look at Datasets for Visuo-Motor Pre-Training

---

### Author Response · Authors · 2024-11-27
**Global response part 1**

Thanks to all reviewers for the valuable feedback.
Here we address points that were common to two or more reviewers. We respond to points raised by individuals in separate responses.

## 1.1 Pre-training loss vs. downstream task performance

Several reviewers queried the core premise of our work -- that pre-training loss is a useful thing to analyze and optimize for. Instead they requested a focus on online performance.

We begin by presenting two new pieces of evidence showing that pre-training loss is closely related to model quality. These have been incorporated into the paper (currently Appendix E).

- 1. __World model evidence.__ We use metrics commonly used to assess the quality of the world models, e.g. [3], originally developed in the video generation literature. Conditioned on an initial real frame and a sequence of real actions, we compare the observations generated by the world model, with the real sequence of observations, measuring FVD, LPIPS and PSNR. Specifically, we generate 1024 videos of 10 seconds each. We perform this for various checkpoints on each model size of our WM-Token-256. This allows a plot of the checkpoint pre-training loss vs world model quality metric to be assessed (e.g. https://ibb.co/gScQQ2S). We find correlations of 0.77, -0.57, 0.83 for LPIPS, PSNR, FVD respectively. This evidences the strong relationship between pre-training loss and world model quality.
- 2. __BC evidence.__ We conducted a meta-analysis of [1] to quantify the strength of relationship between pre-training loss and online return. This is possible in their set up for two reasons. 1) They use a simple environment that could be run cheaply as a gym environment, so all checkpoints could be evaluated. 2) Their data generating policy was an expert rules-based policy, so BC necessarily leads to an expert agent. By taking data points from their Figure 5a and b, we created a plot of pre-training loss vs. online return. This comes out with a correlation coefficient of -0.97 (https://ibb.co/wL0C4kq). This provides strong evidence that in the infinite data regime, pre-training loss is a good proxy for online performance.

In general, we agree with reviewers that when online performance can be measured, this is preferable to pre-training loss.
However, scaling law analyses  require evaluating a large number of checkpoints over multiple training runs. This is not usually feasible for two reasons.
1) For complex environments and applications, such as real robotics or modern video games (our work), measuring online performance is costly. Testing a single BC policy (following finetuning) in the Bleeding Edge environment in our work requires around half a day. Testing the ability to plan using a world model is even more expensive.
2) When pre-training datasets contain a mixture of skill-levels (our work) it is not clear that pre-trained models are 'out-the-box' aligned with high-skill behavior. Hence the name __pre-training__ stage -- models require further fine-tuning or steering to elicit the desired behaviors. This further increases the cost of assessing online performance.

Given these reasons making online performance broadly inaccessible, we now outline qualitative arguments for why pre-training loss __is__ a valuable metric to focus on.

- 1. __Pre-training loss has proven valuable in LLM research.__ LLMs face a similar problem to embodied AI -- one cares about metrics such as factual correctness, reasoning capability, and user engagement, not pre-training loss. Yet these are expensive to measure, and the LLM community has made much progress by focusing on the optimization of the clean intermediate signal of pre-training loss. Why should this not hold for the embodied AI community?
- 2. __Next-token prediction intuition.__ To improve at next-token prediction in world modeling and BC, models intuitively must know more about the environment and behaviors. In BC, better predicting a human's next action requires a model to better understand things about the game objectives the human's are trying to complete, the skill level of individual trajectories, and a variety of alternative behaviors that humans may choose to perform. All these things should create a better pre-trained checkpoint for specialization to a particular downstream task.
- 3. __Validation loss vs. infinite data.__ Previous work has sometimes reported a imperfect relationship between validation loss and performance, e.g. [2]. However, such studies have been performed on datasets of limited size, with held-out validation sets, finding that some amount of over-fitting might be beneficial. Our work is conducted in the infinite data regime, where over-fitting is not possible, so previous insights may not be applicable.

[1] Scaling Laws for Imitation Learning in Single-Agent Games
[2] Hyperparameter Selection for Imitation Learning

---

### Author Response · Authors · 2024-11-27
**Global response part 2**

## 1.2 Increased task diversity

Some reviewers wished to know whether the patterns we have observed in our video game dataset of human behavior would hold in new environments and datasets. Inspired by this, we include a new set of world modeling experiments on the robotics RT-1 dataset (Appendix D), finding that scaling laws again emerge in this set up. Furthermore, we provide further evidence for our claim that the tokenizers compression rate (number of tokens per observation) affects scaling coefficients.

We trained a set of VQVAEs with $V_o = 4096$ and $z_o \in [16, 36, 64, 100, 256]$, for 40,000 updates on batches of 128 (reconstructions here: https://ibb.co/JrgBkKW).
We then train a range of WM-Token model sizes $N \in [0.08M, 0.2M, 0.28M, 0.54M, 0.99M]$ on each VQVAE (e.g. https://ibb.co/9WNFKBV), and measure scaling coefficients using the frontier fit method.

First, we observe that scaling laws similar to those in our main set of experiments emerge, with predictably decreasing losses and similar optimal parameter scaling coefficients.

Secondly, by measuring $a$ where $N_\text{optimal} \propto C^a$ for each VQVAE, repeated three times, we find an that the optimal parameter scaling coefficient increases with number of tokens per observation (https://ibb.co/HNrzkms). This supports our claim that a higher compression rate leads to a lower optimal parameter coefficient.


## 1.3 Comparisons with prior work

Tuyls et al. [1] provide a valuable complimentary analysis to our own work. Here we emphasize several differences.

- 1. Half of our contributions focus on scaling laws for __world modeling__, Tuyls et al. conducted no investigation of this.
- 2. For our BC contributions, Tuyls et al. focused on scaling the width of CNN and LSTM models on datasets created by fixed policies. We consider transformer architectures on datasets of human behavior. We focus on understanding how architectural choices effect scaling coefficients -- surprisingly finding that they heavily influence optimal scaling strategy.
- 3. Section 8 of Tuyls et al. directly mentions that conducting scaling analyses on human datasets is an important direction for future work -- human datasets come with a broad distribution coverage and variation in skill level. This is exactly what our work focuses on.

## References

[1] Scaling Laws for Imitation Learning in Single-Agent Games
[2] Hyperparameter Selection for Imitation Learning
[3] Learning Interactive Real-World Simulators

---

### Meta-Review · Area_Chair_AiSB · 2024-12-24

**Metareview:**

This paper investigates scaling laws in embodied AI, focusing on behavior cloning (BC) and world modeling (WM) tasks. Using a large dataset from gameplay in Bleeding Edge, the study examines the interplay of model size, dataset size, and compute on pre-training losses. The authors demonstrates that scaling laws, similar to those found in language modeling, also apply to embodied AI tasks, revealing distinct trade-offs influenced by tokenization, task type, and architecture. Key findings include how token compression impacts the optimal balance between model and dataset size in WM tasks, and the difficulty of observing scaling laws in BC tasks under modest compute budgets. The aim is to infer insights into optimizng training setups and extending our understanding of scaling in embodied learning tasks.

Reviewers appreciated the novel exploration of scaling laws in embodied AI, noting scaling laws known from LLMs similarly applies to world modeling and BC. They highlighted the impact of tokenization compression on scaling, showing how increasing tokens per image shifts optimal scaling in WM tasks to favor larger models. The paper’s thorough analyses, systematic experimentation, and clear methodologies, including frontier and parametric fits, was also commended for supporting credibility and offering a framework to study trade-offs between model size and computational resources.

However, the reviewers raised concerns about the comprehensiveness of the evaluations for the paper’s claims. They noted the study’s reliance on a single dataset ("Bleeding Edge") and the lack of validation on real-world datasets, which limits its generalizability. Reviewers also questioned the use of pre-training loss as a proxy for downstream performance and called for direct evaluations on practical embodied AI tasks. Concerns were also raised about the relatively smaller-scale models used in experiments and insufficient justification for key hyperparametrs. In the rebuttal, the authors added world modeling results on RT-1 and conducted a metaanalysis on the pretraining loss vis-a-vis downstream task performance, along with providing significant additional details in Appendix D and E. While these efforts partially addressed concerns, in post-rebuttal discussion, it was noted that a broader evaluation across tasks and datasets is still needed for the technical claims of the work.

The AC concurs with the unanimous recommendation of the reviewers and finds that a fresh round of reviews for a more comprehensive draft would be necessary.

**Additional Comments On Reviewer Discussion:**

In the rebuttal, the authors added world modeling results on RT-1 and conducted a metaanalysis on the pretraining loss vis-a-vis downstream task performance, along with providing significant additional details in Appendix D and E. While these efforts partially addressed concerns, in post-rebuttal discussion, it was noted that a broader evaluation across tasks and datasets is still needed for the technical claims of the work.

Initial ratings: 3, 3, 5, 5

Reviewer `QYZ4` increased rating from 3 -> 5.

Final ratings: 3, 5, 5, 5

---

### Decision · Program_Chairs · 2025-01-22

Reject